# Comparative efficacy and safety of statin and fibrate monotherapy: A systematic review and meta-analysis of head-to-head randomized controlled trials

**Joseph E. Blais** [1], **Gloria Kin Yi Tong** [2], **Swathi Pathadka** [1], **Michael Mok** [3], **Ian C. K. Wong** [1,4]*, **Esther W. Chan** [1‡]*

1 Centre for Safe Medication Practice and Research, Department of Pharmacology and Pharmacy, Li Ka Shing Faculty of Medicine, The University of Hong Kong, Hong Kong SAR, China, 2 Pfizer Upjohn Hong Kong Limited, Hong Kong SAR, China, 3 Department of Cardiology, University Hospital Geelong and Deakin University, Geelong, Victoria, Australia, 4 Research Department of Practice and Policy, UCL School of Pharmacy, London, United Kingdom

‡ EWC and ICKW are joint senior authors.
* ewchan@hku.hk (EWC); wongick@hku.hk (ICKW)

**Data Availability Statement:** All relevant data are within the paper and its Supporting information files.

## Abstract

### Objective

To assess whether in adults with dyslipidemia, statins reduce cardiovascular events, mortality, and adverse effects when compared to fibrates.

### Methods

Systematic review and meta-analysis of head-to-head randomized trials of statin and fibrate monotherapy. MEDLINE, EMBASE, Cochrane, WHO International Controlled Trials Registry Platform, and ClinicalTrials.gov were searched through October 30, 2019. Trials that had a follow-up of at least 28 days, and reported mortality or a cardiovascular outcome of interest were eligible for inclusion. Efficacy outcomes were cardiovascular mortality and major cardiovascular events. Safety outcomes included myalgia, serious adverse effects, elevated serum creatinine, and elevated serum alanine aminotransferase. Odds ratios (OR) and 95% confidence intervals (CI) were estimated using the Mantel-Haenszel fixed-effect model, and heterogeneity was assessed using the $I^2$ statistic.

### Results

We included 19 eligible trials that directly compared statin and fibrate monotherapy and reported mortality or a cardiovascular event. Studies had a limited duration of follow-up (range 10 weeks to 2 years). We did not find any evidence of a difference between statins and fibrates for cardiovascular mortality (OR 2.35, 95% CI 0.94–5.86, $I^2$ = 0%; ten studies, n = 2657; low certainty), major cardiovascular events (OR 1.15, 95% CI 0.80–1.65, $I^2$ = 13%; 19 studies, n = 7619; low certainty), and myalgia (OR 1.32, 95% CI 0.95–1.83, $I^2$ = 0%; ten studies, n = 6090; low certainty). Statins had less serious adverse effects (OR 0.57, 95% CI

**Funding:** The author(s) received no specific funding for this work. JEB is supported by the Hong Kong Research Grants Council as a recipient of the Hong Kong PhD Fellowship Scheme. When this study began, GKYT was both a student at the University of Hong Kong and an employee at Otsuka Pharmaceutical (H.K.) Limited. She is currently employed by Pfizer Upjohn Hong Kong Limited. The funders provided support in the form of salaries for GKYT and JEB, but did not have any additional role in the study design, data collection and analysis, decision to publish, or preparation of the manuscript. The specific roles of these authors are articulated in the 'author contributions' section.

**Competing interests:** The authors have read the journal's policy and have the following competing interests: EWC has received honorarium from the Hospital Authority, research grants from Research Grants Council (RGC, HKSAR), Research Fund Secretariat of the Food and Health Bureau (HMRF, HKSAR), National Natural Science Fund of China, National Health and Medical research Council NHMRC, Australia), Wellcome Trust, Bayer, Bristol-Myers Squibb, Pfizer, Janssen, Amgen, Takeda, and Narcotics Division of the Security Bureau of HKSAR, outside the submitted work. ICKW has received research grants from Research Grants Council (RGC, Hong Kong), Innovative Medicines Initiative (IMI), Shire, Janssen-Cilag, Eli-Lily, Pfizer, Bayer, Amgen, and grants from European Union FP7 program, outside the submitted work. GKYT is an employee of Pfizer Upjohn Hong Kong Limited. This does not alter our adherence to PLOS ONE policies on sharing data and materials. There are no patents, products in development or marketed products associated with this research to declare. The other authors have no competing interests to declare.

0.36–0.91, $I^2$ = 0%; nine studies, n = 3749; moderate certainty), less elevations in serum creatinine (OR 0.17, 95% CI 0.08–0.36, $I^2$ = 0%; six studies, n = 2553; high certainty), and more elevations in alanine aminotransferase (OR 1.43, 95% CI 1.03–1.99, $I^2$ = 44%; seven studies, n = 5225; low certainty).

## Conclusions

The eligible randomized trials of statins versus fibrates were designed to assess short-term lipid outcomes, making it difficult to have certainty about the direct comparative effect on cardiovascular outcomes and mortality. With the exception of myalgia, use of a statin appeared to have a lower incidence of adverse effects compared to use of a fibrate.

## Introduction

Statins are the recommended first-line class of lipid-lowering drugs for the primary and secondary prevention of cardiovascular events. Fibrates are used by patients with and without atherosclerotic cardiovascular disease not using statins [1, 2], and are a cost-effective choice for hypercholesterolemia, or mixed dyslipidemia in patients with contraindications or intolerance to statins. Globally, fibrates remain the most used class of nonstatin lipid-lowering drugs and their overall consumption has remained stable between 2008–2018 [3].

Despite being prescribed for decades prior to the introduction of statins, it is not clear as to what extent fibrates have been directly compared to statins for the prevention of cardiovascular events and mortality. To date, systematic reviews of statins and fibrates have made indirect comparisons, usually contrasting the intervention of interest with placebo or usual care [4–8], and have excluded studies of head-to-head comparisons [4–9]. Direct comparisons of therapies are preferable, as indirect comparisons may overestimate the magnitude of treatment differences and decrease confidence in the pooled results [10, 11]. In this study, we aimed to directly assess the efficacy of statin and fibrate monotherapy in adults with dyslipidemia. Our secondary objective was to assess the comparative tolerability and safety of statins and fibrates in the eligible head-to-head studies.

## Methods

We conducted a systematic review and meta-analysis of randomized controlled trials according to the methods recommended by the Cochrane Collaboration [12]. The study protocol is available at www.pharma.hku.hk/research/centre-for-safe-medication-practice-and-research (S1 Protocol). We combined previously published data, therefore, ethics approval was not required.

### Study eligibility criteria

We included randomized controlled trials which directly compared statin monotherapy to fibrate monotherapy in adults with dyslipidemia and that reported mortality or a cardiovascular outcome of interest. Studies that enrolled participants less than 18 years of age or had a follow-up duration of less than 28 days were excluded. No language restrictions were applied.

### Outcomes of interest

**Primary outcome:**

- Cardiovascular mortality

**Secondary outcomes:**

- Efficacy

  - All-cause mortality

  - Major cardiovascular events, defined as a composite of cardiovascular death, coronary artery disease, myocardial infarction, unstable angina, and stroke

  - Individual components of major cardiovascular events

- Safety outcomes included adverse effects associated with the use of statins or fibrates [13, 14]:

  - Muscle-related adverse effects included rhabdomyolysis, elevations in creatine kinase (CK), and myalgia

  - Elevations in serum alanine aminotransferase (ALT)

  - Renal adverse events, included kidney injury and elevations in serum creatinine

  - Participant withdrawal due to adverse effects

  - Number of serious adverse effects

  - New diagnosis or worsening of diabetes mellitus

  - Non-cardiovascular mortality

  - Venous thromboembolism

**Exploratory outcomes included the percent reduction from baseline to end of study for the following lipid concentrations:**

- Total cholesterol (TC)

- Low-density lipoprotein cholesterol (LDL-C)

- High-density lipoprotein cholesterol (HDL-C)

- Non-high-density lipoprotein cholesterol (non-HDL-C)

- Triglycerides

- Apolipoprotein B (apoB)

Elevations in laboratory outcomes (CK, ALT, and serum creatinine), were defined according to the lowest clinically meaningful threshold as reported by study investigators. Myalgia was defined as the number of participants described as having myalgia or muscle pain. If myalgia or muscle pain was not reported, then we extracted the number of cases with musculoskeletal pain. Because most included studies were designed to assess reductions in cholesterol concentrations, post-hoc we also assessed the surrogate efficacy outcomes of reductions in lipid levels.

## Search methods for identification of studies

We systematically searched for published and unpublished studies using Ovid MEDLINE and Epub Ahead of Print, In-Process & Other Non-Indexed Citations and Daily, EMBASE via Ovid, the Cochrane Central Register of Controlled Trials, ClinicalTrials.gov, and the WHO International Controlled Trials Registry Platform, from database inception until October 30, 2019. Search strategies were developed using keywords and medical subject headings for statins, fibrates, and the key efficacy and safety outcomes. We used recommended search terms

(filters) which provide the best balance of sensitivity and specificity for studies of treatment [15, 16]. The complete search strategy is described in S1 File.

Two authors (GKYT and JEB) independently screened study abstracts and titles. Relevant full-text articles were then retrieved and assessed independently (GKYT and JEB) for inclusion according to a standard list of exclusion criteria that were applied in priority sequence (Table 1 in S1 File). Discrepancies were resolved by consensus.

### Data extraction

Two authors (GKYT and SP or SP and JEB) independently extracted relevant study characteristics and outcomes using a standardized data extraction form. Data were extracted from all identified relevant study reports. If discrepancies between published journal articles and trial registries were identified, we extracted results from ClinicalTrials.gov, since reporting of outcomes and severe adverse events is more complete than in journal publications [17, 18].

### Risk of bias assessment

Two authors (SP and JEB) independently assessed the risk of bias within each study using the Cochrane Risk of Bias Tool [19]. For appraisal of performance and detection bias, we broadly grouped outcomes into subjective and objective outcomes.

### Data synthesis and statistical analysis

Study level characteristics were pooled and are reported as percentages, or means and standard deviations (SD). Studies with double-zeros, meaning the outcome was not reported or had zero events in both the statin and fibrate arms, were excluded from the selected meta-analysis models, and for single-zero studies, a 0.5 continuity correction was added as the default software setting [20]. After receiving reviewer reports regarding our analysis method for rare events, we changed our primary analysis model to the Mantel-Haenszel (MH) fixed-effect model to estimate odds ratios (OR) for dichotomous outcomes. For continuous lipid outcomes, we used an inverse variance fixed-effect model to estimate mean differences (MD). When we could not extract or calculate a standard deviation for the continuous outcomes, for each treatment group, we imputed the largest standard deviation reported from the included studies for that treatment group [12].

To examine the consistency of our findings, we undertook several subgroup and sensitivity analyses. The first subgroups were primary prevention, secondary prevention, and unreported baseline prevalence of cardiovascular disease. Primary prevention studies were arbitrarily defined as those which enrolled participants with a baseline history of cardiovascular disease of 10% or less, while studies with more than 10% of participants with baseline cardiovascular disease were defined as secondary prevention [4]. We also analyzed studies according to the primary type of dyslipidemia (primary hypercholesterolemia, mixed dyslipidemia, and other), according to fibrate drug, and whether the study included participants with diabetes (pre-specified as > 90% of study participants with a baseline history of diabetes mellitus).

We also undertook sensitivity analyses for dichotomous outcomes by estimating Peto OR, and our original analysis–MH random-effects model for risk ratios (S1 Table). The Peto method may be appropriate and unbiased for the analysis of rare events when the event is rare (<1%), the groups are balanced, and the intervention effects are small [21]. In addition, it may be the least biased method in the presence of a true treatment effect when studies with double-zeroes are excluded [22]. For continuous outcomes, we estimated MD using a random-effects model. To show the effect of imputing standard deviations for the continuous outcomes, we separated studies requiring imputation of standard deviations into a subgroup in the forest

plots. Heterogeneity across studies was assessed using the $I^2$ statistic. For outcomes reported in 10 or more studies, reporting bias was assessed visually using funnel plots. All statistical analyses were conducted in Review Manager Version 5.3 [20].

## Assessment of the certainty of the evidence

The GRADE approach (grading of recommendations assessment, development, and evaluation) was used to assess the certainty of the evidence [11]. Outcomes were categorized into high, moderate, low, and very low certainty of evidence. We imported data from Review Manager into GRADEpro GDT software to create a summary of evidence table for the outcomes judged to be most important (cardiovascular mortality, all-cause mortality, major cardiovascular events, study withdrawal due to adverse effects, serious adverse effects, myalgia, elevated ALT, and elevated serum creatinine) [23].

# Results

## Description of studies and patient population

Details of the study selection process are shown in Fig 1. After screening titles and abstracts, 244 articles were excluded since they did not report mortality or a cardiovascular event of interest. Five records were identified by reviewing systematic reviews and reference lists of the included studies. The characteristics of the included studies are available in Table 1. Nineteen studies (reported in 24 articles) met the inclusion criteria [24–42], and allocated a total of 7619 participants to either statin or fibrate monotherapy, totaling approximately 4745 person-years of follow-up. Five studies [31, 33, 36–38] had a follow-up duration of 24 weeks or longer, and the longest follow-up was two years [38]. Only one study reported the outcome of cardiovascular events as a primary or secondary outcome of interest [36]; the remainder reported mortality and cardiovascular events as adverse events. Eight studies allowed upward dose titration of statins [24, 25, 31, 33, 35, 39, 40, 42], while all studies assigned participants to a fixed dose of fibrate. Four studies were conducted in primary prevention [30, 32–34], 10 studies assessed secondary prevention [25–27, 31, 35–38, 40, 42], and five studies [24, 28, 29, 39, 41] did not report the number of participants with a baseline diagnosis of cardiovascular disease. Earlier studies tended to enroll participants with primary hypercholesterolemia, while more recent studies enrolled participants with mixed dyslipidemia. Only one study met our pre-specified definition of enrolling participants with diabetes mellitus, thus we did not conduct this subgroup analysis [39].

## Risk of bias

Results of the risk of bias assessment across studies and for each study is shown in Figs 1, 2 in S1 File. Most studies did not report allocation concealment and were judged to be at either unclear risk of bias or at high risk of bias. Other reasons for high risk of bias were blinding of participants and personnel and blinding of outcome assessment for subjective outcomes. Four studies were judged to be at high risk of bias because of incomplete outcome data, which was due to higher participant withdrawals in the fibrate group [30, 31], or excessive dropouts in both treatment groups [33, 36]. Reporting bias was assessed graphically using funnel plots for the outcomes of cardiovascular mortality, all-cause mortality, major cardiovascular events, myocardial infarction, withdrawal due to adverse events, myalgia, and elevated CK (Figs 3–9 in S1 File). We strongly suspected reporting bias only for the outcome of participant withdrawal due to adverse events.

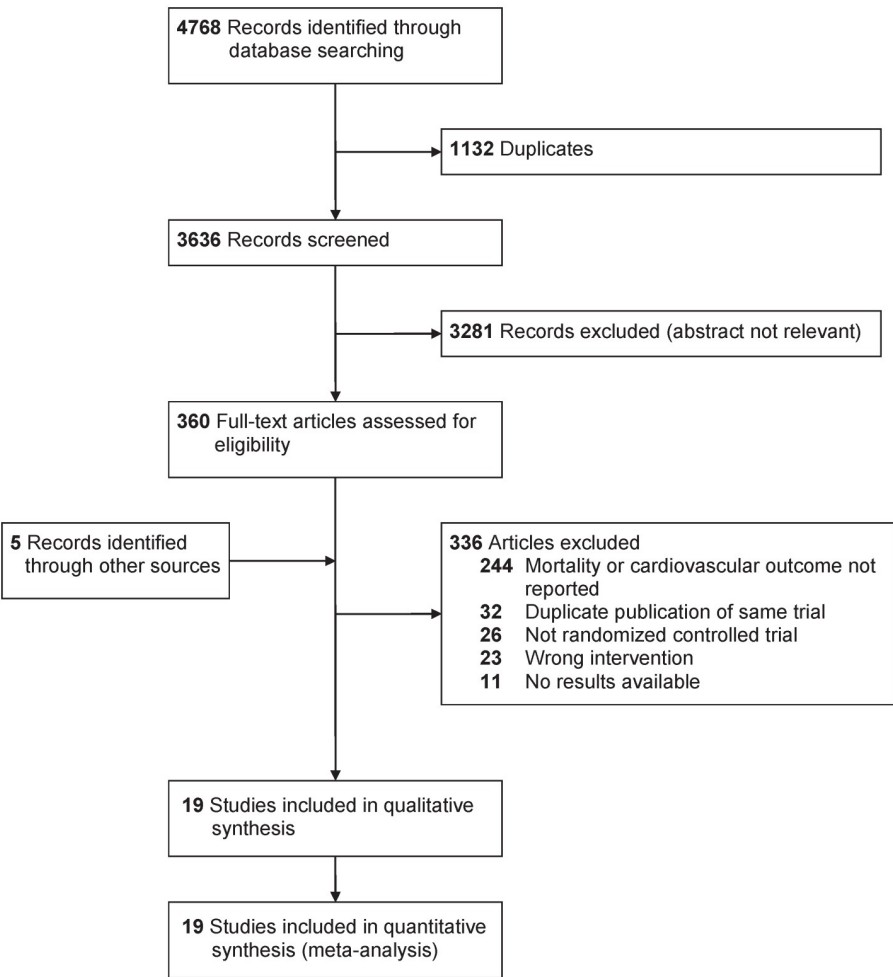

**Fig 1. PRISMA (preferred reporting items for systematic reviews and meta-analyses) flowchart of head-to-head studies evaluating statin and fibrate monotherapy on mortality and cardiovascular outcomes.**

## Certainty of the evidence

A summary of evidence table for the key outcomes of interest is included in S1 File. The certainty in our estimates for each outcome ranged from very low (unstable angina) to high (elevated serum creatinine). Because many studies had a short duration of follow-up and low numbers of events, all clinical efficacy outcomes and several safety outcomes were judged to have serious imprecision. Surrogate lipid outcomes (percent reduction in TC, LDL-C, HDL-C, non-HDL-C, triglycerides, and apoB) were downgraded for indirectness (surrogate outcomes for cardiovascular events) resulting in moderate certainty of evidence.

## Efficacy outcomes

**Cardiovascular mortality.** A total of 11 studies reported at least one cardiovascular death. We excluded Pover 1995 from the estimation of cardiovascular mortality, since the authors reported that 80% of all deaths in the study were due to acute myocardial infarction, but did not report these deaths according to treatment assignment [33]. The limited number of events resulted in imprecise estimates and there was no evidence of a difference in cardiovascular

**Table 1. Summary of the included study characteristics directly comparing statin and fibrate monotherapy.**

| Study | Region(s) | Number of participants | Follow-up duration (weeks) | Dyslipidemia | Lipid test inclusion criteria (mmol/L) | Mean baseline lipid levels (mmol/L) | Primary or secondary prevention (CVD %) | Mean age (SD) | Diabetes (%) | Men (%) | Statin | Fibrate | Funding |
|---|---|---|---|---|---|---|---|---|---|---|---|---|---|
| Tikkanen 1988 | Finland | 334 | 12 | Primary hypercholesterolemia | TC > 6.22 · TG ≤ 3.95 | TC: 8.9 · TG: 2.0 · LDL-C: 6.7 · HDL-C: 1.3 | Secondary (CAD: 43%) | 51.6 (NR) | NR | 49% | Lovastatin 20–80 mg daily | Gemfibrozil 600 mg twice daily | Merck, Sharp & Dohme |
| Ziegler 1990 | France · Switzerland | 184 | 10 | Primary hypercholesterolemia | TC ≥ 7.78 · LDL-C ≥ 5.05 · TG < 3.95 | TC: 10.0 · TG: 1.5 · LDL-C: 8.1 · HDL-C: 1.1 | Secondary (CAD: 43%) | 46.0 (NR) | NR | 70% | Simvastatin 20–40 mg daily | Fenofibrate 200 mg twice daily | NR |
| Arntz 1991 | Germany | 96 | 12 | Primary hypercholesterolemia | TG < 3.95 | TC: 9.4 · TG: 1.9 · LDL-C: 7.4 · HDL-C: 1.3 | NR | 52.5 (11.7) | 0% | 50% | Pravastatin 20–40 mg daily | Bezafibrate 400 mg daily | NR |
| Sanchez 1991 | Spain | 39 | 12 | Primary hypercholesterolemia | TC ≥ 6.50 · TG ≤ 3.95 | TC: 7.8 · TG: 1.6 · LDL-C: 5.8 · HDL-C: 1.2 | Secondary (CVD: 72%) | 57.5 (9.0) | 0% | 62% | Lovastatin 20–80 mg daily | Bezafibrate 200 mg three times daily | Merck, Sharp & Dohme |
| D'Agostino 1992 | United States | 104 | 18 | Primary hypercholesterolemia | TC ≥ 6.21 or LDL-C ≥ 4.92 · TC ≥ 6.21 or LDL-C ≥ 4.14 (with definite CAD or 2 CAD risk factors) | TC: 7.5 · TG: 1.9 · LDL-C: 5.5 · HDL-C: 1.2 | Secondary (CAD: 24%) | 55.0 (13.0) | 1% | 58% | Lovastatin 20–40 mg daily | Gemfibrozil 600 mg twice daily | Merck, Sharp & Dohme |
| Wiklund 1993 | Sweden · Finland | 143 | 12 | Primary hypercholesterolemia | TC ≥ 6.00 · TG < 4.00 | TC: 7.3 · TG: 1.8 · LDL-C: 5.2 · HDL-C: 1.2 | NR | 54.1 (NR) | NR | 69% | Pravastatin 40 mg daily | Gemfibrozil 600 mg twice daily | Swedish Medical Research Council, Swedish Heart and Lung Foundation, King Gustav V and Queen Victoria Foundation, Bristol-Myers Squibb Company |
| Greten 1994 | Germany | 131 | 12 | Primary hypercholesterolemia | LDL-C ≥ 4.10 · TG ≤ 3.40 | TC: 7.3 · TG: 1.8 · LDL-C: 5.2 · HDL-C: 1.2 | NR | 52.4 (NR) | 1% | 44% | Fluvastatin 40 mg daily | Bezafibrate 400 mg daily | Sandoz AG Nurnberg |

(Continued)

**Table 1.** (Continued)

| Study | Region(s) | Number of participants | Follow-up duration (weeks) | Dyslipidemia | Lipid test inclusion criteria (mmol/L) | Mean baseline lipid levels (mmol/L) | Primary or secondary prevention (CVD %) | Mean age (SD) | Diabetes (%) | Men (%) | Statin | Fibrate | Funding |
|---|---|---|---|---|---|---|---|---|---|---|---|---|---|
| Lintott 1995 | New Zealand | 129 | 24 | Moderate hypercholesterolemia | TC 5.20 to 7.80 <br> LDL-C ≥ 4.14 <br> TG ≤ 4.60 | TC: 6.8 <br> TG: 1.8 <br> LDL-C: 4.9 <br> HDL-C: 1.1 | Secondary (CAD: 51%) | 56.0 (9.4) | NR | 62% | Simvastatin 5–10 mg daily Pravastatin 10 mg daily | Bezafibrate 200 mg three times daily | Merck, Sharp & Dohme |
| Sinzinger 1995 | Austria | 524 | 24 | Moderate hypercholesterolemia | TC 5.20 to 7.80 (no lipid drug therapy) or 6.50 to 7.80 (on lipid drug therapy) | TC: 7.1 <br> TG: 2.4 <br> LDL-C: 4.8 <br> HDL-C: 1.3 | Secondary (Angina: 18%) | 55.0 (10.5) | 8% | 45% | Lovastatin 20 mg daily | Bezafibrate 400 mg daily | Boehringer Mannheim Austria |
| Sweany 1995 | United States, Austria, Germany, Brazil, New Zealand | 168 | 18 | Primary hypercholesterolemia | LDL-C ≥ 4.90m (no risk factor) or | TC: 6.9 <br> TG: 2.3 <br> LDL-C: 4.7 <br> HDL-C: 1.2 | NR | 58.5 (NR) | 100% | 43% | Simvastatin 10–40 mg daily | Gemfibrozil 600 mg twice daily | Merck Research Laboratories |
| Power 1995 | United Kingdom | 2467 | 52 | Primary hypercholesterolemia | LDL-C ≥ 4.10 (one or more risk factors according to NCEP 1998 guidelines) <br> TC > 7.80 <br> or > 6.50 and presence of two CV risk factors <br> TG 1.00 to 5.70 (or < 5.70 mmol/L if already treated with a fibrate) | TC: 8.2 <br> TG: 2.5 <br> LDL-C: 5.8 <br> HDL-C: 1.2 | Primary (CHD: 5%) | 54.9 (NR) | 1% | 55% | Pravastatin 20–40 mg daily | Gemfibrozil 600 mg twice daily Bezafibrate 400 mg daily | Bristol-Myers Squibb |
| De Lorgeril 1999 | France | 64 | 12 | Primary hypercholesterolemia | TC > 6.50 <br> TG ≤ 3.50 | TC: 7.2 <br> TG: 2.0 <br> LDL-C: 5.1 <br> HDL-C: 1.2 | Secondary (CAD: 100%) | 54.1 (NR) | NR | 100% | Simvastatin 20 mg daily | Fenofibrate 200 mg daily | Laboratoire Fournier, Daix, France |
| Skoloudik 2007 | Czech Republic | 376 | 104 | Hypercholesterolemia | TC > 5.00 and < 8.00 <br> TG ≤ 5.00 | TC: 6.5 <br> TG: 2.2 <br> LDL-C: 4.2 <br> HDL-C: 1.3 | Secondary (CAD: 27%) | 63.4 (8.5) | 23% | 63% | Fluvastatin 40 mg daily | Fenofibrate 200 mg daily | NR |

(Continued)

**Table 1.** (Continued)

| Study | Region(s) | Number of participants | Follow-up duration (weeks) | Dyslipidemia | Lipid test inclusion criteria (mmol/L) | Mean baseline lipid levels (mmol/L) | Primary or secondary prevention (CVD %) | Mean age (SD) | Diabetes (%) | Men (%) | Statin | Fibrate | Funding |
|---|---|---|---|---|---|---|---|---|---|---|---|---|---|
| Goldberg 2009 | North America | 392 | 16 | Mixed dyslipidemia | LDL-C ≥ 3.37; TG ≥ 1.69; HDL-C < 1.02 (men) or HDL-C < 1.28 (women) | TC: 6.8; TG: 3.2; LDL-C: 4.2; HDL-C: 1.0 | NR | 54.8 (10.9) | 24% | 50% | Atorvastatin 20–80 mg daily | Fenofibrate 135 mg daily | Abbott |
| Jones 2009 | North America | 922 | 16 | Mixed dyslipidemia | LDL-C ≥ 3.37; TG ≥ 1.69; HDL-C < 1.02 (men) or HDL-C < 1.28 (women) | TC: 7.8; TG: 1.6; LDL-C: 5.8; HDL-C: 1.2 | Primary (CAD: 8%) | 55.1 (10.5) | 20% | 49% | Rosuvastatin 10–40 mg daily | Fenofibrate 135 mg daily | Abbott |
| Mohiuddin 2009 | North America | 419 | 16 | Mixed dyslipidemia | LDL-C ≥ 3.37; TG ≥ 1.69; HDL-C < 1.02 (men) or HDL-C < 1.28 (women) | TC: 6.7; TG: 3.2; LDL-C: 4.1; HDL-C: 1.0 | Primary (CAD: 7%) | 54.2 (10.3) | 23% | 48% | Simvastatin 20–80 mg daily | Fenofibrate 135 mg daily | Abbott |
| Roth 2010 | United States | 507 | 16 | Mixed dyslipidemia | LDL-C ≥ 3.37; TG ≥ 1.69; HDL-C < 1.02 (men) or HDL-C < 1.28 (women) | TC: 6.7; TG: 2.7; LDL-C: 3.9; HDL-C: 1.1 | Primary (CAD: 6%) | 54.8 (11.0) | 28% | 42% | Rosuvastatin 5 mg daily | Fenofibrate 135 mg daily | Abbott and Astra Zeneca |
| Sano 2010 | Japan | 274 | 52 | Elevated remnant lipoprotein levels | RLP-C ≥ 5.0 mg/dL; TC ≥ 4.66 and < 6.73; TG ≥ 1.69 and < 4.52 | TC: 5.5; TG: 2.3; LDL-C: 3.3; HDL-C: 1.1 | Secondary (CAD: 100%) | 56.0 (9.4) | 34% | 88% | Pravastatin 10–20 mg daily | Bezafibrate 200–400 mg daily | Ministry of Education, Culture, Sports, Science and Technology, Health and Labor Sciences Research Grants for Comprehensive Research on Aging and Health, Japan |
| Foucher 2015 | Czech Republic; Mexico; Argentina; Poland; Romania; Russian Federation; Germany | 346 | 12 | Mixed dyslipidemia | LDL-C 1.81 to 3.37; TG ≥ 1.69 | TC: 4.7; TG: 2.7; LDL-C: 2.7; HDL-C: 1.2 | Secondary (CVD: 44%) | 60.1 (9.1) | 58% | 65% | Simvastatin 20–40 mg daily | Fenofibrate 145 mg daily | Abbott |

Abbreviations: CAD, coronary artery disease; CHD, coronary heart disease; CVD, cardiovascular disease; HDL-C, high-density lipoprotein cholesterol; LDL-C, low-density lipoprotein cholesterol; NCEP, National Cholesterol Education Program; NR, not reported; RLP-C, remnant-like lipoprotein particles cholesterol; TC, total cholesterol; TG, triglycerides.

To convert LDL-C, HDL-C, or total cholesterol to mg/dL divide by 0.0259. To convert triglycerides to mg/dL divide by 0.0113.

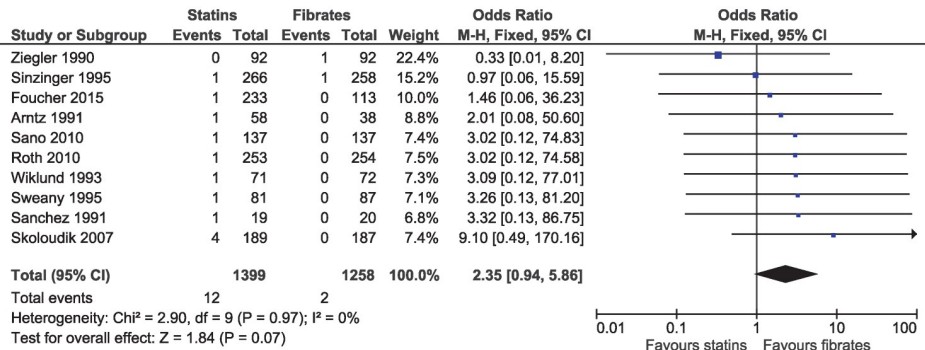

**Fig 2. Forest plot of comparison: Statins versus fibrates, outcome: Cardiovascular mortality.**

mortality between statins and fibrates (OR 2.35, 95% CI 0.94–5.86, $I^2$ = 0%; ten studies, n = 2657; low certainty; Fig 2).

**All-cause mortality.** Eleven studies with follow-up ranging from 10 weeks to 2 years, reported a death from any cause (OR 1.67, 95% CI 0.87–3.22, $I^2$ = 0%; n = 5124; Fig 3). Evidence was downgraded for imprecision and high or uncertain risk of bias resulting in a low certainty of evidence.

**Major cardiovascular events.** In the eligible studies, no evidence of a difference for statins and fibrates was observed for major cardiovascular events (OR 1.15, 95% CI 0.80–1.65, $I^2$ = 13%; 19 studies, n = 7619; low certainty; Fig 4). Individual cardiovascular outcomes were rare resulting in imprecise estimates, and there was no evidence of a difference for myocardial infarction (OR 0.78, 95% CI 0.49–1.24, $I^2$ = 0%; 15 studies, n = 6362; Fig 5), coronary artery disease (OR 0.98, 95% CI 0.34–2.78, $I^2$ = 0%; six studies, n = 2505; Fig 6), unstable angina (OR 2.38, 95% CI 0.90–6.24, $I^2$ = 52%; four studies, n = 1200; Fig 7), and stroke (OR 2.04, 95% CI 0.86–4.82, $I^2$ = 36%; three studies, n = 1157; Fig 8).

## Surrogate efficacy outcomes

There were greater reductions in percent change from baseline for TC (MD -11.49%, 95% CI -12.20 to -10.77, $I^2$ = 96%; 15 studies, n = 6002; S1 Fig), LDL-C (MD -19.63%, 95% CI -20.70

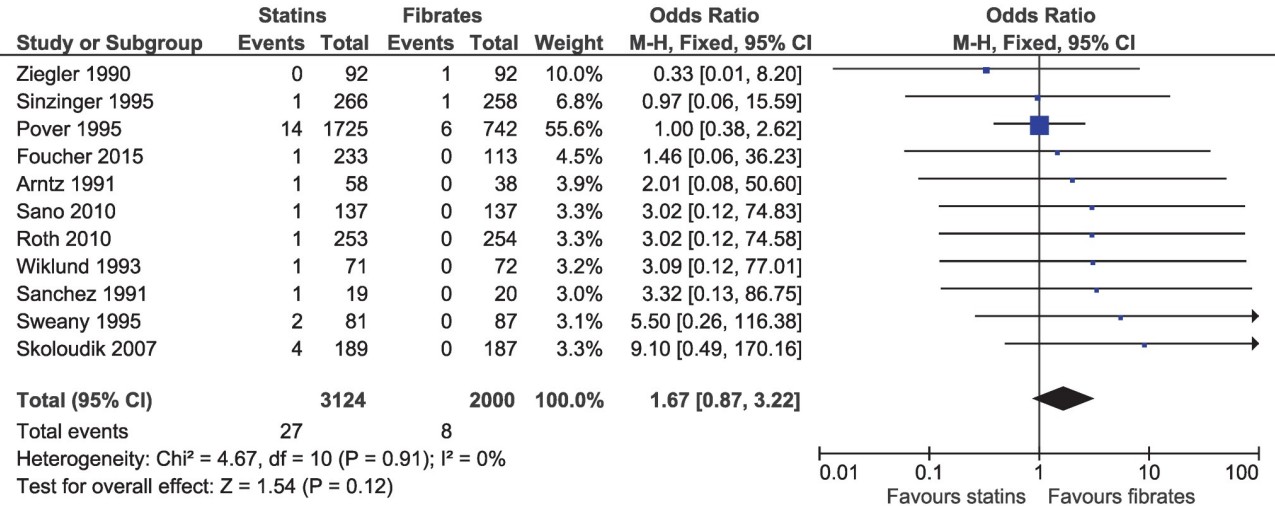

**Fig 3. Forest plot of comparison: Statins versus fibrates, outcome: All-cause mortality.**

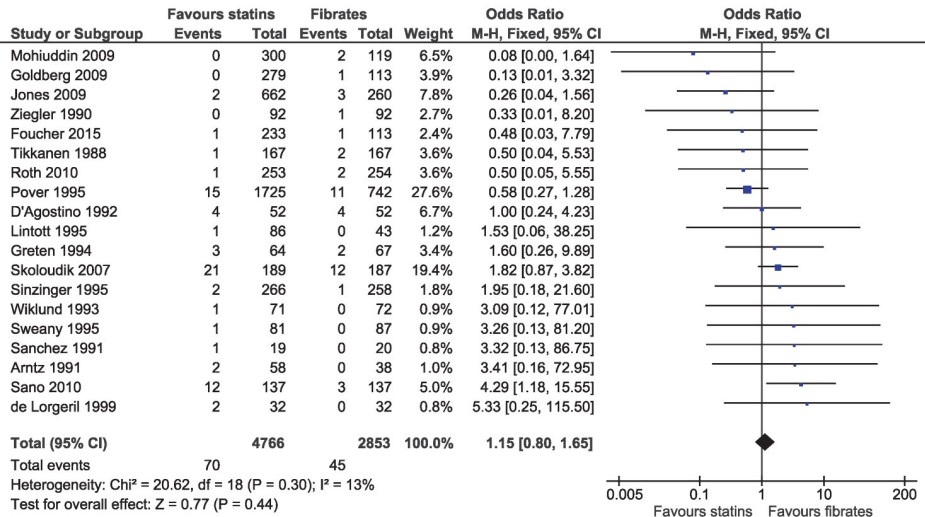

**Fig 4. Forest plot of comparison: Statins versus fibrates, outcome: Major cardiovascular events.**

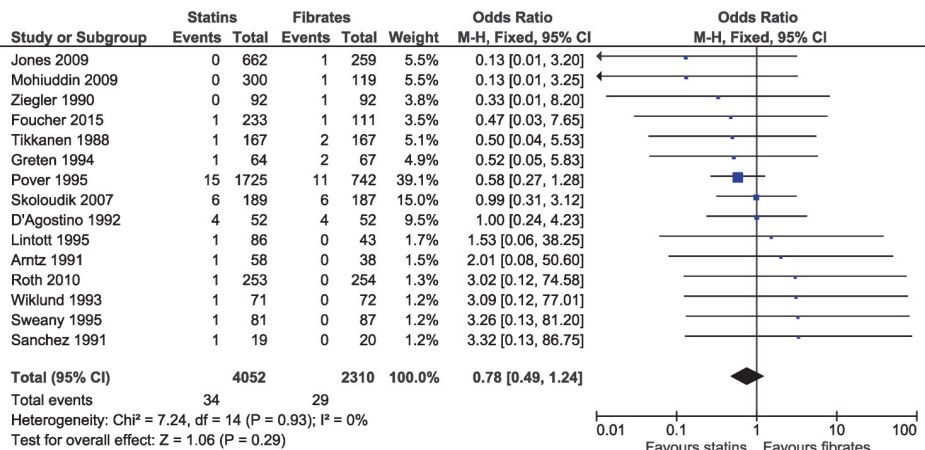

**Fig 5. Forest plot of comparison: Statins versus fibrates, outcome: Myocardial infarction.**

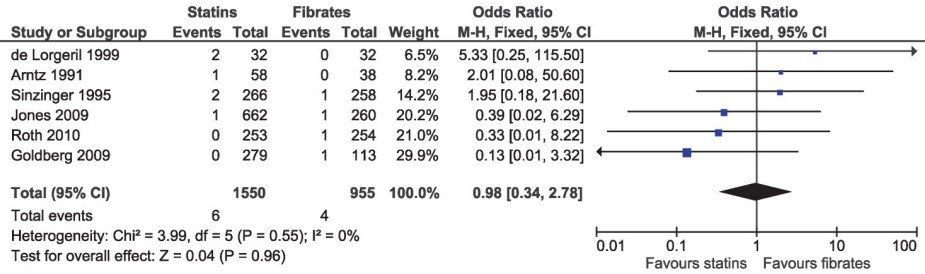

**Fig 6. Forest plot of comparison: Statins versus fibrates, outcome: Coronary artery disease.**

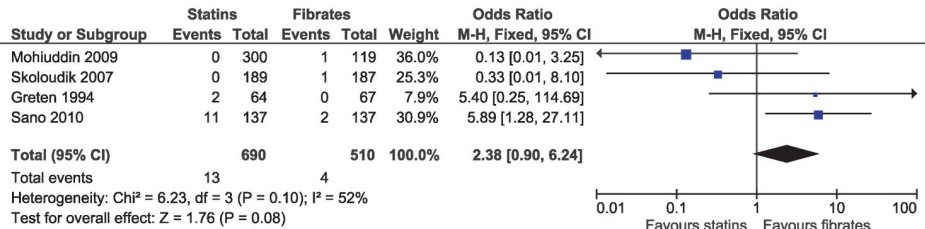

**Fig 7. Forest plot of comparison: Statins versus fibrates, outcome: Unstable angina.**

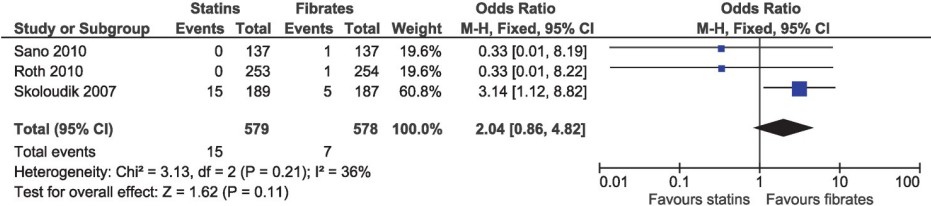

**Fig 8. Forest plot of comparison: Statins versus fibrates, outcome: Stroke.**

to -18.55, $I^2$ = 96%; 15 studies, n = 5795; S2 Fig), non-HDL-C (MD -20.94%, -22.46 to -19.41, $I^2$ = 93%; four studies, n = 2008; S3 Fig), and apoB (MD -16.83%, 95% CI -18.10 to -15.56, $I^2$ = 86%; nine studies, n = 3003; S4 Fig) among statin therapy than fibrate therapy. Fibrates reduced triglyceride levels by 15.34% (95% CI 13.52 to 17.15, $I^2$ = 71%; 15 studies, n = 5922; S5 Fig) and increased HDL-C concentrations (MD 8.15%, 95% CI 9.23 to 7.07, $I^2$ = 69%; 15 studies, n = 5850; S6 Fig) more than statins. Pooled results for studies which we imputed SD were of similar magnitude, but tended to be closer to the null (except for triglyceride levels) as compared to studies with no SD imputation.

## Safety outcomes

**Tolerability.** Statins were associated with a lower risk of study withdrawal due to adverse effects (OR 0.71, 95% CI 0.55–0.93, $I^2$ = 4%; 16 studies, n = 4680; low certainty; Fig 9) and

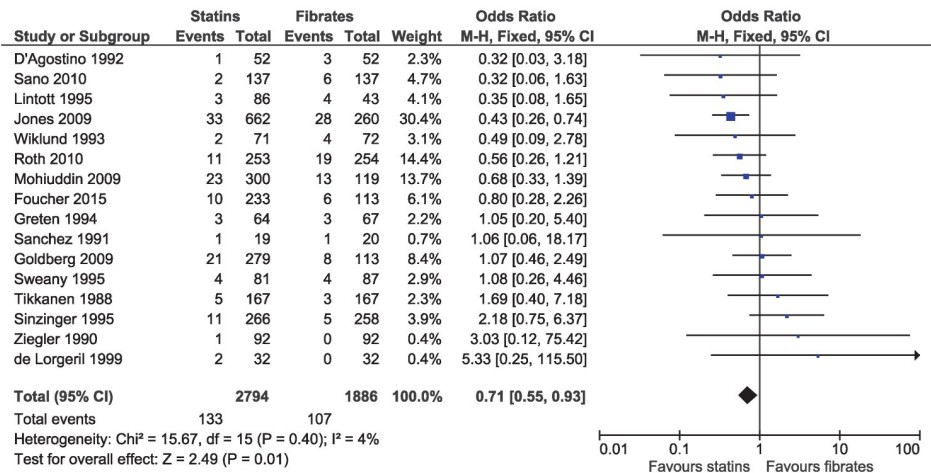

**Fig 9. Forest plot of comparison: Statins versus fibrates, outcome: Study withdrawal due to adverse effects.**

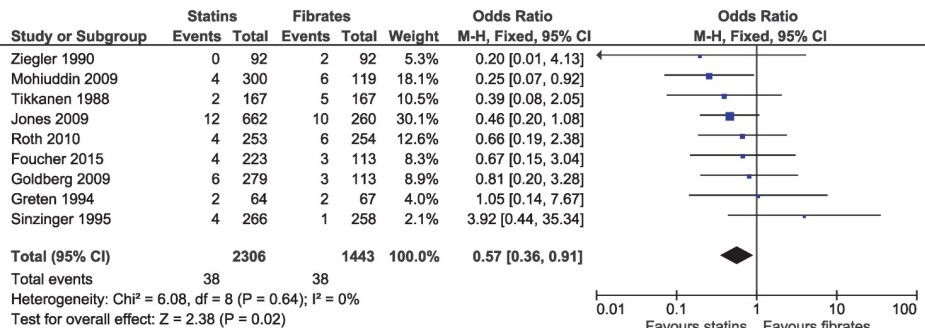

**Fig 10. Forest plot of comparison: Statins versus fibrates, outcome: Serious adverse effects.**

serious adverse effects (OR 0.57, 95% CI 0.36–0.91, $I^2$ = 0%; nine studies, n = 3749; moderate certainty; Fig 10). Six studies included in the outcome of withdrawal due to adverse effects permitted dose increases in statin treatment at specified intervals [25, 31, 35, 39, 40, 42]. Dose titration may influence the reported adverse effects as participants initiated or maintained on lower doses of statins may be less likely to experience adverse effects, as compared to those initiated on a standard dose of fibrates and was considered in the risk of bias assessment.

**Muscle-related adverse effects.** There was no clear evidence of a difference for myalgia (OR 1.32, 95% CI 0.95–1.83, $I^2$ = 0%; ten studies, n = 6090; low certainty; Fig 11) or for elevations in CK (OR 1.43, 95% CI 0.99–2.06, $I^2$ = 0%; 14 studies, n = 6762; S7 Fig). No study reported rhabdomyolysis in participants receiving statin or fibrate monotherapy.

**Hepatic and renal adverse effects.** In the primary analysis, statins increased the risk of elevated ALT (OR 1.43, 95% CI 1.03–1.99, $I^2$ = 44%; seven studies, n = 5225; low certainty; Fig 12). Statins greatly reduced the risk of elevated serum creatinine (OR 0.17, 95% CI 0.08–0.36, $I^2$ = 0%; six studies, n = 2553; high certainty; Fig 13). Because of the small number of events and different outcome descriptions, we could not pool studies for the outcome of kidney injury. Three studies reported kidney injury related outcomes as renal failure [30, 43], renal impairment [27], or renal dysfunction [36]. A total of four cases of kidney injury were reported in the fibrate group and zero in the statin group.

## Other outcomes

A limited number of events did not permit pooling of studies for the outcomes of new onset diabetes mellitus, venous thromboembolism, and non-cardiovascular mortality. Two studies

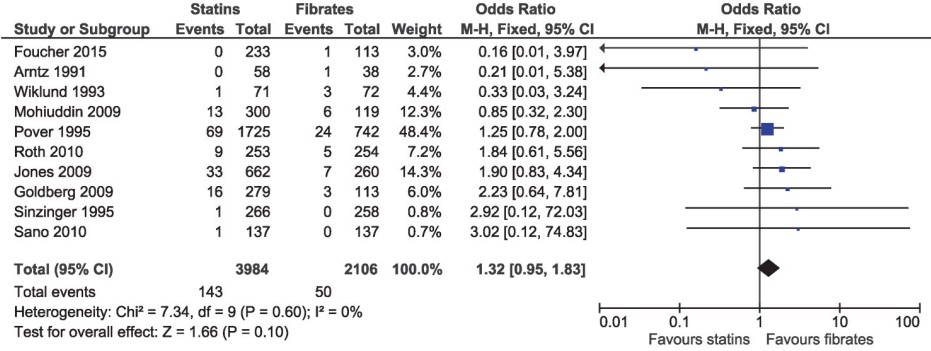

**Fig 11. Forest plot of comparison: Statins versus fibrates, outcome: Myalgia.**

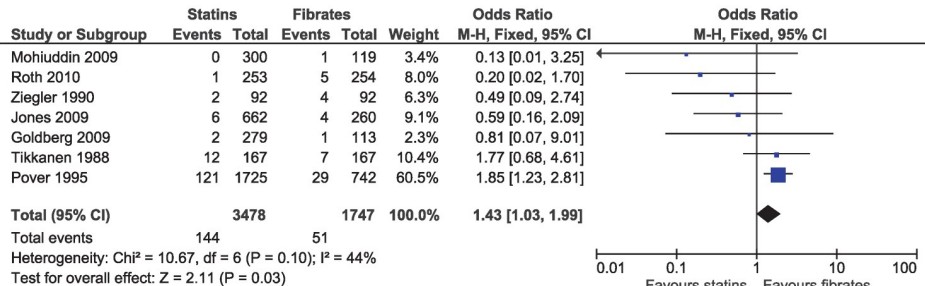

**Fig 12. Forest plot of comparison: Statins versus fibrates, outcome: Elevated alanine aminotransferase.**

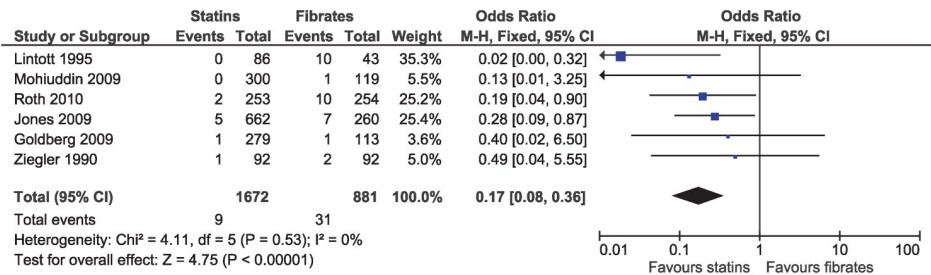

**Fig 13. Forest plot of comparison: Statins versus fibrates, outcome: Elevated serum creatinine.**

[39, 42] reported the outcome of increased fasting blood glucose, with one case occurring in a participant assigned to fibrate [39] and the other in a participant assigned to statin therapy [42]. In a third study, one case each of diabetes mellitus, decompensated type 2 diabetes mellitus, and inadequate control of diabetes mellitus were reported (two participants in the statin group and one participant in the fibrate group) [27]. A single case of deep vein thrombosis was reported in the fibrate group [28, 44]. Non-cardiovascular deaths were reported in two studies [33, 39].

## Subgroup and sensitivity analyses

The results of subgroup analyses and are mostly consistent with the main analyses (S1 File). Evidence of differences between subgroups was apparent for major cardiovascular events and elevated ALT. Statins appeared to reduce the risk of major cardiovascular events in the primary prevention and mixed dyslipidemia subgroups, while fibrates reduced the risk in the secondary prevention and other dyslipidemia subgroup (Fig 12 in S1 File). A high degree of heterogeneity between studies for elevated ALT resulted in different estimates by fibrate drug (Fig 17 in S1 File).

The results of sensitivity analyses are shown in S1 Table. Cardiovascular mortality, all-cause mortality, stroke, unstable angina, and elevated ALT were all sensitive to model choice and should be considered as hypothesis generating results. Estimates using the Peto method were generally consistent with the main analysis, despite not always meeting all three suggested criteria for analysis of rare events [21].

## Discussion

Although fibrates have been available for decades prior to statins, no large-scale clinical trial with a follow-up duration longer than two years has directly compared the effects of statins

versus fibrates for the clinical outcomes of death or cardiovascular events. After an extensive literature search, this systematic review included 19 randomized controlled trials of head-to-head comparisons of statin and fibrate monotherapy. Nonetheless, nearly all the included studies were not designed to assess the effects of statins and fibrates on important efficacy outcomes. However, despite the limitations regarding the assessment of efficacy, the results indicate that statins are probably less likely to cause adverse effects than fibrates.

This study has implications for clinicians since about 5% of patients, who do not take statins, are using a fibrate [1, 2]. While fibrate monotherapy remains an evidence-based treatment option for adults with dyslipidemia, this study adds further direct evidence to support the role of statins as a potentially safer treatment with respect to tolerability, a reduction in serious adverse effects, and a reduced risk of elevated serum creatinine levels. The subgroup analyses of participants with mixed dyslipidemia also show that statins are likely a more effective treatment. This is an interesting finding since fibrates, when compared with placebo, have been shown in at least two meta-analyses to benefit patients with mixed dyslipidemia [45, 46].

At least two systematic reviews using indirect comparisons demonstrate that statins and fibrates produce a similar magnitude of reduction in major vascular events. The first investigated the association between non-HDL-C lowering and major vascular events, with similar estimates for statins (risk ratio (RR) 0.80, 95% CI 0.77 to 0.82) and fibrates (RR 0.79, 85% CI 0.71 to 0.88) per 1-mmol/L reduction in non-HDL-C [47]. A second systematic review investigated various interventions to reduce LDL-C and also found significant associations with statins (RR 0.80, 95% CI 0.78 to 0.82) and fibrates (RR 0.88, 95% CI 0.83 to 0.93) for the outcome of major vascular events [8]. The lack of evidence of a difference between statins and fibrates in this study is consistent with the similar relative benefit observed in indirect comparisons. However, it is more likely a result of the short duration of follow-up and small number of outcome events, since the time to benefit in large-scale trials of lipid-lowering medications is typically greater than one year [48]. Therefore, detecting a difference in efficacy between active treatments, if one exists, would require a much larger sample size and a longer duration of follow-up than is available from the eligible trials.

A strength of this study is the assessment of several adverse effects which are frequently not included or briefly addressed in systematic reviews of lipid-lowering drugs [7, 49–51]. The result demonstrating that fibrates increase serum creatinine levels are consistent with at least three previous systematic reviews of fibrates [5, 9, 52]. For the outcome of elevated ALT, our study included a greater number of studies and outcome events than both Cochrane reviews of fibrates [4, 9], improving the directness of our estimates. However, it is difficult to be very confident in our results for elevated ALT as this outcome varied in both subgroup and sensitivity analyses.

Muscle symptoms are well-known adverse effects associated with both statin and fibrate treatment. To assess comparative muscle safety, we included both muscle symptoms (myalgia) and the objective outcome of elevated CK (which can often be asymptomatic). Our included studies identified a greater number of total myalgia events than other comprehensive fibrate reviews [6, 9]. Despite including a larger number of myalgia events in this study, our estimates for myalgia remained uncertain, with the bounds of the confidence interval including both a potential reduction and an increase in risk. Determining the true incidence and risk factors for statin associated muscle-related adverse effects remains an active area of research.

## Strengths and limitations

This study has several strengths. First, we included trials of head-to-head comparisons of statins and fibrates that have not usually been included in previous systematic reviews. Second, a

careful review of full-text articles by two independent authors, allowed us to identify all cardio-vascular event outcomes for study inclusion. However, this study is limited by the eligible randomized controlled trials. The short duration of follow-up and rare events resulted in reduced power to detect differences between groups, and some estimates were sensitive to the choice of meta-analysis model and should be considered as hypothesis generating. Lastly, in the context of secondary prevention patients, the choice of statin drug and statin dose intensity in many of the eligible trials does not align with current guideline recommendations for treatment with high-intensity statins.

### Future research on statins versus fibrates

Given the more robust body of evidence for statins in reducing the risk of cardiovascular events and an improved safety profile, further research is needed to understand the role of fibrates for the prevention of cardiovascular events. We identified one ongoing study, the Pita-vastatin or Bezafibrate Intervention, Assessment of Antiarteriosclerotic Effect study (PIO-NEER), that is directly comparing pitavastatin to bezafibrate monotherapy with the primary outcome of change in mean carotid intima-media thickness [53]. The results of PIONEER may add additional information to the potential role of fibrate monotherapy particularly in patients with both elevated LDL-C and triglyceride concentrations. In addition, observational studies with adequate sample size and long-term follow-up can complement the available data from randomized controlled trials.

## Conclusions

Direct comparisons of statins and fibrates are available from randomized controlled trials of adults with primary dyslipidemia, mixed dyslipidemia and moderate hypercholesterolemia. The eligible trial evidence focused on surrogate lipid outcomes and no evidence of a difference was found for statins and fibrates for cardiovascular events, or cardiovascular mortality. Estimates for clinical efficacy outcomes are severely limited by a short duration of follow-up, risk of bias, and imprecision. Apart from muscle-related adverse effects, statins appear to have an improved safety profile which supports their current role as the preferred treatment option for the prevention of cardiovascular events.

## Supporting information

**S1 File. Search strategy, exclusion criteria, risk of bias assessment, reporting bias, summary of findings, and forest plots for subgroup analyses.**
(DOCX)

**S1 Checklist. PRISMA checklist.**
(DOC)

**S1 Protocol. Study protocol.**
(PDF)

**S1 Table. Main analysis and sensitivity analyses for statistical models.**
(XLSX)

**S1 Fig. Forest plot of comparison: Statins versus fibrates, outcome: Total cholesterol.**
(PDF)

**S2 Fig. Forest plot of comparison: Statins versus fibrates, outcome: Low-density lipoprotein cholesterol.**
(PDF)

**S3 Fig. Forest plot of comparison: Statins versus fibrates, outcome: Non-high-density lipoprotein cholesterol.**
(PDF)

**S4 Fig. Forest plot of comparison: Statins versus fibrates, outcome: Apolipoprotein B.**
(PDF)

**S5 Fig. Forest plot of comparison: Statins versus fibrates, outcome: Triglycerides.**
(PDF)

**S6 Fig. Forest plot of comparison: Statins versus fibrates, outcome: High-density lipoprotein cholesterol.**
(PDF)

**S7 Fig. Forest plot of comparison: Statins versus fibrates, outcome: Elevated creatine kinase.**
(PDF)

## Acknowledgments

We thank Miriam Haendler and Barbora Schonfeldova for translating articles.

## Author Contributions

**Conceptualization:** Joseph E. Blais, Gloria Kin Yi Tong, Esther W. Chan.

**Formal analysis:** Joseph E. Blais, Gloria Kin Yi Tong.

**Investigation:** Gloria Kin Yi Tong, Swathi Pathadka.

**Project administration:** Joseph E. Blais, Gloria Kin Yi Tong.

**Resources:** Ian C. K. Wong, Esther W. Chan.

**Supervision:** Ian C. K. Wong, Esther W. Chan.

**Validation:** Gloria Kin Yi Tong, Swathi Pathadka.

**Visualization:** Joseph E. Blais, Gloria Kin Yi Tong.

**Writing – original draft:** Joseph E. Blais.

**Writing – review & editing:** Gloria Kin Yi Tong, Swathi Pathadka, Michael Mok, Ian C. K. Wong, Esther W. Chan.

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
