## [Decision Letter · Decision Letter 0]

7 Dec 2020

PONE-D-20-24944

Comparative efficacy and safety of statin and fibrate monotherapy: A systematic review and meta-analysis of head-to-head randomized controlled trials

PLOS ONE

Dear Dr. Blais,

Thank you for submitting your manuscript to PLOS ONE. After careful consideration, we feel that it has merit but does not fully meet PLOS ONE’s publication criteria as it currently stands. Therefore, we invite you to submit a revised version of the manuscript that addresses the points raised during the review process.

Please respond to each of the points raised by the editor and 2 reviewers below. 

We look forward to receiving your revised manuscript.

Kind regards,

James M Wright

Academic Editor

PLOS ONE

Additional Editor Comments:

For lines 304-305 remove the negative signs as it could confuse readers.  

In the Discussion creatinine is misspelled as creatine in at least 2 places.  

Journal Requirements:

"JEB is supported by the Hong Kong Research Grants Council as a recipient of the Hong Kong PhD Fellowship Scheme."

"I have read the journal's policy and the authors of this manuscript have the following competing interests: EWC has received honorarium from the Hospital Authority, research grants from Research Grants Council (RGC, HKSAR), Research Fund Secretariat of the Food and Health Bureau (HMRF, HKSAR), National Natural Science Fund of China, National Health and Medical research Council NHMRC, Australia), Wellcome Trust, Bayer, Bristol-Myers Squibb, Pfizer, Janssen, Amgen, Takeda, and Narcotics Division of the Security Bureau of HKSAR, outside the submitted work. ICKW has received research grants from Research Grants Council (RGC, Hong Kong), Innovative Medicines Initiative (IMI), Shire, Janssen-Cilag, Eli-Lily, Pfizer, Bayer, Amgen, and grants from European Union FP7 program, outside the submitted work. KYT is an employee of Pfizer Upjohn Hong Kong Limited. The other authors have no competing interests"

We note that one or more of the authors are employed by a commercial company: Pfizer Upjohn Hong Kong Limited.

3.1. Please provide an amended Funding Statement declaring this commercial affiliation, as well as a statement regarding the Role of Funders in your study. If the funding organization did not play a role in the study design, data collection and analysis, decision to publish, or preparation of the manuscript and only provided financial support in the form of authors' salaries and/or research materials, please review your statements relating to the author contributions, and ensure you have specifically and accurately indicated the role(s) that these authors had in your study. You can update author roles in the Author Contributions section of the online submission form.

3.2. Please also provide an updated Competing Interests Statement declaring this commercial affiliation along with any other relevant declarations relating to employment, consultancy, patents, products in development, or marketed products, etc.  

Reviewers' comments:

Reviewer's Responses to Questions

**Comments to the Author**

1. Is the manuscript technically sound, and do the data support the conclusions?

Reviewer #1: Partly

Reviewer #2: Partly

2. Has the statistical analysis been performed appropriately and rigorously? 

Reviewer #1: Yes

Reviewer #2: Yes

3. Have the authors made all data underlying the findings in their manuscript fully available?

Reviewer #1: Yes

Reviewer #2: Yes

4. Is the manuscript presented in an intelligible fashion and written in standard English?

Reviewer #1: Yes

Reviewer #2: Yes

5. Review Comments to the Author

Reviewer #1: 1. Is the manuscript technically sound, and do the data support the conclusions? Statins versus fibrates OR of 0.17 should report the results as a decrease not an elevation of serum creatinine due to statins

Reviewer #2: Statins are first-line agents for hypercholesterolemia. Current guidelines do not recommend the use of fibrates as alternative to statins in the absence of hypertriglyceridemia. However, fibrates have the labeled indication of hypercholesterolemia or mixed dyslipidemia when statins are contraindicated or not tolerated. Therefore, the relative efficacy of fibrates versus statins in clinically relevant outcomes is an interesting question.

The review uses the Cochrane method. After looking for randomized controlled trials that directly compare statins with fibrates, authors find very little evidence about their relative efficacy. Evidence for safety is more robust, favoring statins in most outcomes.

Methods are sound but the reporting has to be improved. Some statements do not correspond with the findings. Definition of secondary does not correspond with clinical practice. There are some inconsistence in the assessment of the quality of evidence. Some data need revision. Authors should rewrite some sentences to avoid confusion. Discussion has to be improved. There are some mistakes.

The article needs several changes prior to being published.

Major issues

Statements that are not based in findings

Line 45 “…but might be associated with an increased risk of myalgia (OR 1.32, 95% CI 0.95–1.83…” since there is no statistically significant difference in the risk of myalgia, the sentence has to be rewritten to say that there was no clear evidence to support a difference in the risk of myalgia.

Line 330. “ the relevance of elevations in CK, which tended to favour fibrate monotherapy (OR 1.43, 95% CI 0.99–2.06,” is not an evidence-based statement; the confidence interval includes no difference.

Line 39. “We did not find any eligible evidence of a difference”

As some evidence has been selected, it would be more appropriate to say

… not find evidence of a difference…

Lines 66-68. This is a more accurate statement  "To date, systematic reviews of fibrates have made indirect comparisons, usually contrasting the intervention of interest with placebo or usual care and have excluded head-to head comparisons [5-10] "

Definition that does not correspond with clinical practice

Line 173. “studies with more than 10% of participants with baseline cardiovascular disease were defined as secondary prevention”. This definition is misleading. A study with only 18% of patients with baseline cardiovascular disease are not equivalent to a secondary prevention population in which all patients have cardiovascular disease. The authors should change this designation.

Inconsistence in the assessment of the quality of evidence

The certainty of evidence for cardiovascular mortality is downgraded for allocation concealment, incomplete outcome data, blinding of outcome assessment, and selective reporting in several studies. It is not clear why these problems do not affect others outcomes. Moreover, the study Pover 1995, with high risk of bias on these items, is excluded from the analysis of cardiovascular mortality and included in those of all-cause mortality and major cardiovascular events.

Line 238. A reason for high risk of bias is blinding of outcome assessment for subjective outcomes. Myalgia is the more subjective outcome. In spite of most of the events coming from studies with high risk of bias, evidence was rated as moderate. Downgrading of evidence must be consistent.

Confusing sentences

Line 44. “.. and elevations in serum creatinine (OR 0.17…” is confusing. It should be with reduced risk of elevation in serum creatinine..

Line 47. As there is a dichotomous outcome it would be better to say: ..and might be associated with an increased risk of elevation in alanine aminotransferase

“Primary prevention studies were arbitrarily defined as those which enrolled participants with a baseline history of cardiovascular disease of 10% or less,” this phrase needs rewording

Line 278. “Pooling eligible studies suggests statins and fibrates for major cardiovascular events”  does not make sense.  I think you mean ....are not different in their effect on major cardiovascular events" 

Line 188. “To assess the validity of imputing standard deviations for the continuous outcomes, we separated studies requiring imputation of standard deviations into a subgroup.” The validity of imputing standard deviations is assumed. It is not clear how they can be assessed in this way. Presenting as subgroups can be a way to inform the reader in what studies the imputation was done.

In the forest plots, it is not clear why SD are different in studies to which SD have been imputed.

Data that need review

Line 62. The source of unpublished data should be reported.

Line 185. Ref 22 does not fit with the statement.

Line 187. I cannot find information about the Peto method in GRADEpro [23].

Table 1.

Sinzinger 1995: It is not clear how the 18% of participants with CAD disease has been calculated. The article only provided data of per-protocol population. It seems to be 109 patients with CAD disease (myocardial infarction, angina or coronary bypass/PTCA)

Goldberg 2009, Jones 2009, Mohiuddin 2009, Roth 2010: Treatment period is 12 weeks

Fig 1. Flowchart: there is a mistake. The branch “5 records identified through other sources” has to be higher than “360 full-text..”. It would be noted 19 studies (coming from 24 articles); otherwise, numbers are inconsistent.

Line 272. Results of all-cause mortality in the text and in Fig 3 are different.

Line 299. Results of LDL-C in the text and in S2Fig are different.

In Sano 2010, the number of major cardiovascular events is 12 in the statin group.

Issues with Discussion

Line 421: “Our included studies identified a greater number of total myalgia events

than other comprehensive fibrate reviews.[7, 10] This may be because ascertainment of muscle related adverse effects may have been more deliberate in our included studies as these are known adverse effects of statins and fibrates, and adverse effects may have been assessed more systematically than in larger placebo controlled studies.” This is very speculative. Others reviews have a different scope and include different studies with placebo groups.

The PROMINENT study do not compare directly statins with fibrates, so it is not relevant for the discussion.

Line 404. “The short duration of study follow-up, and perhaps a similar anticipated reduction in cardiovascular events, explain why we could not detect a difference in clinical efficacy outcomes”. Author should clarify this statement. Does it mean that the results are consistent with those of the previous indirect comparisons?

In order to put the review in the context of current clinical practice, it would be helpful to note that statins and doses used in the included trials are not of high intensity treatment currently recommended for secondary prevention.

Minor issues

The authors should remark that the technical name of statins is HMG CoA reductase inhibitors.

Line 179. Finding only one study that met the definition of diabetes is a result.

Line 58-60: I would like the authors clarify the period in which the use of fenofibrate has been evaluated. References 1-4 provide quite old data. Is there current information?

Line 72. “In addition, we examined tolerability and safety outcomes for the included studies”. Tolerability and safety outcomes have to be analysed according to the methodology. Re-write the sentence.

The assessment of safety is critical in a systematic review about medicines. The author should re-write the sentence.

6. PLOS authors have the option to publish the peer review history of their article (what does this mean?). If published, this will include your full peer review and any attached files.

Reviewer #1: No

Reviewer #2: **Yes: **Javier Garjon

---

## [Author Response · Author response to Decision Letter 0]

15 Jan 2021

Please see our attached word document which contains the detailed response to reviewers.

---

## [Editor Report · Decision Letter 1]

18 Jan 2021

PONE-D-20-24944R1

Comparative efficacy and safety of statin and fibrate monotherapy: A systematic review and meta-analysis of head-to-head randomized controlled trials

PLOS ONE

Dear Joseph Blais,

Thank you for re-submitting your manuscript to PLOS ONE. After careful consideration, we feel that it has merit but does not fully meet PLOS ONE’s publication criteria as it currently stands. Therefore, we invite you to submit a revised version of the manuscript that addresses the points raised during the review process.

In your re-submission the main issue is that the abstract (the most important and most read part of the paper) needs improvement.  I have made suggested copy-edits in the abstract below.  If that works for you we should be able to proceed with publication.  

We look forward to receiving your revised manuscript.

Kind regards,

James M Wright

Academic Editor

PLOS ONE

Additional Editor Comments (if provided):

Objective To assess whether in adults with dyslipidemia, statins reduce cardiovascular events, mortality, and adverse effects when compared to fibrates.

Methods Systematic review and meta-analysis of head-to-head randomized trials of

statin and fibrate monotherapy. MEDLINE, EMBASE, Cochrane, WHO International

Controlled Trials Registry Platform, and ClinicalTrials.gov were searched through

October 30, 2019. Trials that had a follow-up of at least 28 days, and reported mortality

or a cardiovascular outcome of interest were eligible for inclusion. Efficacy outcomes

were cardiovascular mortality and major cardiovascular events. Safety outcomes

included myalgia, serious adverse effects, elevated serum creatinine, and elevated

serum alanine aminotransferase. Odds ratios (OR) and 95% confidence intervals (CI)

were estimated using the Mantel-Haenszel fixed-effect model, and heterogeneity was

assessed using the I 2 statistic.

Results We included 19 eligible trials that directly compared statin and fibrate

monotherapy and reported mortality or a cardiovascular event. Studies had a limited

duration of follow-up (range 10 weeks to 2 years). We did not find any evidence

of a difference between statins and fibrates for cardiovascular mortality (OR 2.35, 95% CI

0.94–5.86, I 2 =0%; ten studies, n=2657; low certainty), major cardiovascular events

(OR 1.15, 95% CI 0.80–1.65, I 2 =13%; 19 studies, n=7619; low certainty), and

myalgia (OR 1.32, 95% CI 0.95–1.83, I 2 =0%; ten studies, n=6090; low certainty).

Statins had less serious adverse effects (OR 0.57, 95% CI 0.36–0.91, I 2

=0%; nine studies, n=3749; moderate certainty), less elevations in

serum creatinine (OR 0.17, 95% CI 0.08–0.36, I 2 =0%; six studies, n=2553; high

certainty), and more elevations in alanine aminotransferase (OR

1.43, 95% CI 1.03–1.99, I 2 =44%; seven studies, n=5225; low certainty).

Conclusions The eligible randomized trials of statins versus fibrates were designed to

assess short-term lipid outcomes, making it difficult to have certainty

about the direct comparative effect on cardiovascular outcomes and mortality. With the

exception of myalgia, use of a statin appeared to have a lower incidence of adverse effects compared to use of a fibrate.

---

## [Author Response · Author response to Decision Letter 1]

18 Jan 2021

Dear Dr Wright,

Thank you for taking the time to copy-edit the abstract. We agree with all the editorial suggestions. Changes to the manuscript (only the abstract) are enclosed in a marked-up copy with track changes.

Yours sincerely,

Joseph Blais

On behalf of co-authors

---

## [Editor Report · Decision Letter 2]

20 Jan 2021

Comparative efficacy and safety of statin and fibrate monotherapy: A systematic review and meta-analysis of head-to-head randomized controlled trials

PONE-D-20-24944R2

Dear Dr. Joseph Blais,

We’re pleased to inform you that your manuscript has been judged scientifically suitable for publication and will be formally accepted for publication once it meets all outstanding technical requirements.

Kind regards,

James M Wright

Academic Editor

PLOS ONE
---

## [Editor Report · Acceptance letter]

29 Jan 2021

PONE-D-20-24944R2 

Comparative efficacy and safety of statin and fibrate monotherapy: A systematic review and meta-analysis of head-to-head randomized controlled trials 

Dear Dr. Blais:

I'm pleased to inform you that your manuscript has been deemed suitable for publication in PLOS ONE. Congratulations! Your manuscript is now with our production department. 

Kind regards, 

on behalf of

Professor James M Wright 

Academic Editor

PLOS ONE